# Anatomic versus Low Tibial Tunnel in Double-Bundle Posterior Cruciate Ligament Reconstruction: Clinical and Radiologic Outcomes with a Minimum 2-Year Follow-Up

**DOI:** 10.3390/medicina60040545

**Published:** 2024-03-27

**Authors:** Chung-Yu Chen, Chen-Heng Hsu, Poyu Chen, Kuo-Yao Hsu, Cheng-Pang Yang, Huan Sheu, Shih-Sheng Chang, Chih-Hao Chiu

**Affiliations:** 1Department of Orthopedic Surgery, Linkou Chang Gung Memorial Hospital, Taoyuan 333, Taiwan; mp2287@cgmh.org.tw (C.-Y.C.); mp1796@cgmh.org.tw (C.-H.H.); mp0526@cgmh.org.tw (C.-P.Y.); 2Department of Occupational Therapy and Graduate Institute of Behavioral Sciences, College of Medicine, Chang Gung University, Taoyuan 333, Taiwan; poyuchen@mail.cgu.edu.tw; 3Department of Orthopedic Surgery, New Taipei Municipal Tucheng Hospital, New Taipei City 236, Taiwan; emsequoia@adm.cgmh.org.tw; 4Department of Orthopedic Surgery, Taoyuan Chang Gung Memorial Hospital, Taoyuan 333, Taiwan; mp1333@cgmh.org.tw (H.S.); u8301007@adm.cgmh.org.tw (S.-S.C.); 5Bone and Joint Research Center, Linkou Chang Gung Memorial Hospital, Taoyuan 333, Taiwan; 6Comprehensive Sports Medicine Center (CSMC), Chang Gung Memorial Hospital, Taoyuan 333, Taiwan; 7College of Medicine, Chang Gung University, Taoyuan 333, Taiwan

**Keywords:** knee, posterior cruciate ligament, double-bundle, killer turn, tibial tunnel, side-to-side difference, kneeling stress view

## Abstract

There is currently no consensus on the optimal placement of the tibial tunnel for double-bundle posterior cruciate ligament (PCL) reconstruction. The purpose of this study was to compare the clinical and radiologic outcomes of double-bundle PCL reconstruction utilizing anatomic versus low tibial tunnels. We conducted a retrospective cohort study involving patients who underwent double-bundle PCL reconstruction between Jan 2019 and Jan 2022, with a minimum follow-up of 2 years (*n* = 36). Based on the tibial tunnel position on postoperative computed tomography, patients were categorized into two groups: anatomic placement (group A; *n* = 18) and low tunnel placement (group L; *n* = 18). We compared the range of motion, stability test, complications, and side-to-side differences in tibial posterior translation using kneeling stress radiography between the two groups. There were no significant differences between the groups regarding clinical outcomes or complication rates. No significant differences in the posterior drawer test and side-to-side difference on kneeling stress radiography (2.5 ± 1.2 mm in group A vs. 3.7 ± 2.0 mm in group L; *p* = 0.346). In conclusion, the main findings of this study indicate that both anatomic tunnel and low tibial tunnel placements in double-bundle PCL reconstruction demonstrated comparable and satisfactory clinical and radiologic outcomes, with similar overall complication rates at the 2-year follow-up.

## 1. Introduction

Posterior cruciate ligament (PCL) tears represent significant injuries, with the potential for long-term consequences on the knee joint. These injuries often occur as a result of high-energy trauma, such as motor vehicle or sports-related accidents, with a male predominance and an average age at injury around the third decade of life [1,2]. Despite the fact that PCL insufficiency has a relatively low incidence compared to anterior cruciate ligament (ACL) injury, estimated at approximately 2 per 100,000 persons annually, recent research indicates that conservative treatment approaches may increase the risk of developing knee osteoarthritis. This heightened risk can stem from factors such as severe PCL laxity or the progression of meniscus tears [3,4,5].

With an evolving understanding of the anatomy and biomechanics of the PCL and advances in surgical techniques, arthroscopic PCL reconstruction has become a widely adopted treatment for individuals with PCL insufficiency [6]. However, historical data indicated that the outcomes of PCL reconstruction were less predictable and led to inferior results when compared to ACL reconstruction [7,8]. A systematic review of single-bundle PCL reconstructions demonstrated an overall graft failure rate of 12.5%. Moreover, only 50 to 82% of patients undergoing PCL reconstruction managed to return to their preinjury activity level [9]. However, since Kennedy et al. reported a codominant relationship between the anterolateral bundle (ALB) and the posteromedial bundle (PMB) of the PCL in resisting posterior tibial translation at all flexion angles, there has been an increasing trend towards double-bundle PCL reconstruction [10]. Recently, numerous randomized controlled studies and systematic reviews with meta-analysis have demonstrated that anatomic-based double-bundle PCL reconstruction can restore the native biomechanics of the intact knee closely and yield superior objective outcomes compared to single-bundle reconstruction [11,12,13,14,15].

Meanwhile, the acute angle of the graft at the tibial tunnel exit, commonly referred to as the “killer turn,” has been identified as one of the primary factors contributing to the residual laxity and graft failure of transtibial PCL reconstruction [16,17,18]. To reduce the “killer turn” effect and improve the clinical efficacy of PCL reconstruction, several techniques have been proposed. These include high tibial osteotomy, the modified tibial tunnel technique, which creates a tibial tunnel from the anterior lateral side of the tibia, preserving remnants as a soft tissue cushion, and utilizing inlay or onlay techniques that do not require a tibial tunnel [19,20,21,22,23,24]. Fanelli et al. introduced a technique involving the placement of the tibial tunnel lower in relation to the footprint, situated below the champagne glass drop-off. This innovative approach aimed to mitigate the sharpness of the killer turn by breaking it into two gentler angles [25,26]. Subsequent long-term follow-up studies have reported positive clinical and functional outcomes associated with this technique [27,28].

Nonetheless, to the best of our knowledge, clinical outcomes comparing the anatomic tibial tunnel (above the champagne glass drop-off) and low tibial tunnel (below the champagne glass drop-off) placements in double-bundle PCL reconstruction have not been reported yet. The primary aim of this study was to assess and compare the clinical and radiologic outcomes of double-bundle PCL reconstructions utilizing anatomic and low tibial tunnels. We hypothesized that at the 2-year follow-up, low tibial tunnel placement would have superior outcomes compared to anatomic tibial tunnel placement.

## 2. Materials and Methods

### 2.1. Patient Enrollment and Study Design

After obtaining the approval of ethics committee from the Institutional Review Board for this study, we conducted a retrospective review of patients who underwent double-bundle PCL reconstruction in our hospital between January 2019 and January 2022 for inclusion in this study. The diagnosis of PCL injury was confirmed through physical examination, assessment of side-to-side difference (STSD) in posterior tibial translation using kneeling stress radiography, and magnetic resonance imaging (MRI). Written informed consent was obtained from all individual participants enrolled in this study.

PCL reconstruction was recommended for patients exhibiting more than 8 mm of STSD in posterior tibial translation, as measured through preoperative stress radiography, and in cases where nonoperative treatment had proven ineffective [29]. In our practice, the favored indications for low tibial tunnel placement included young individuals with high activity, lower BMI, or greater preoperative STSD. However, the final decision was made with respect to the preference and consideration of each patient and surgeon. The exclusion criteria were as follows: (1) single-bundle PCL reconstruction, (2) multiligament injury requiring combined ligament surgery, (3) revision PCL reconstruction, (4) fractures around the ipsilateral knee, (5) bilateral PCL injury, and (6) advanced-stage knee osteoarthritis (Kellgren–Lawrence grade 3 and 4). A total of 128 patients were initially assessed for this study. Among them, 41 patients met the aforementioned criteria. Of the 41 patients, 5 patients were excluded due to loss to follow-up or incomplete radiologic data at the 2-year period. Ultimately, 36 patients were enrolled in this study (Figure 1).

### 2.2. Surgical Technique

The arthroscopic double-bundle PCL reconstruction technique was performed with Achilles tendon allografts described as in a previous study [8]. Under the visualization via a posteromedial portal, the PCL tibial footprint was identified and debrided along the PCL facet distally, approximately 10 mm below the champagne glass drop-off. An Acufex PCL tibial guide (Smith & Nephew, Andover, MA, USA) was introduced through the anteromedial portal. For the anatomic tibial tunnel placement, the desired guide pin positions were the inferior and lateral corners of the native PCL tibial footprint. For the low tibial tunnel placement, the target position for the guide pin was 5–10 mm below the champagne glass drop-off, with the same medial-to-lateral dimension as the anatomic tibial tunnel. Subsequently, the guide pin was over-drilled with a 12 mm cannulated reamer to create the tibial tunnel.

To prepare the femoral tunnel, the Acufex PCL femoral guide (Smith & Nephew, Andover, MA, USA) was introduced through the anteromedial portal. The guide pins were inserted in an outside-in fashion. The anterolateral bundle (ALB) was centered between the trochlear and medial arch points, and the posteromedial bundle (PMB) was positioned in front of the medial intercondylar ridge, about 6 mm offset from the cartilage edge [30]. A bone bridge (approximately 8 mm) between the femoral tunnels was ensured before drilling [28]. ALB and PMB femoral tunnels were over-drilled with 9 mm and 8 mm reamers, respectively. Grafts were retrogradely passed from the tibial to the femoral tunnels. Primary fixation used bioabsorbable interference screws (Linvatec BioScrew; ConMed, Largo, FL, USA). Femoral ALB fixation was at 90° knee flexion with maximal anterior tibial translation, and PMB was fixed at 0° knee extension [31]. Tibial backup fixation employed two 4.5 mm cortical screws with washers.

### 2.3. Postoperative Rehabilitation

Following the surgery, PCL dynamic braces were applied immediately to all the patients, exerting dynamically increasing force with increased flexion angle [32]. It was recommended that the brace should be worn at all times for a minimum of 24 weeks. The patients were encouraged to engage in quadriceps isometric contraction training. In the initial 4 weeks post-surgery, partial weight bearing and limited flexion up to 90° were permitted. Full weight bearing and unrestricted range of motion were allowed after 6 weeks. Patients could resume daily activities without the PCL dynamic brace after 6 months, and sports activities were permitted at 12 months post-surgery [33]. Moreover, this protocol remained applicable regardless of intervention for meniscus tears.

### 2.4. Evaluation of Tibial Tunnel Position on 3-Dimensional Computed Tomography (3D-CT)

Postoperative CT scans were performed for all patients within one month post-PCL reconstruction. To categorize patients based on tibial tunnel position, the sagittal view of the proximal tibia and 3D-CT were examined to pinpoint the center of the tibial tunnel. Patients were assigned to either the anatomic group (group A) or the low tunnel group (group L) based on whether the center of the tibial tunnel was located above or below the champagne glass drop-off, respectively (Figure 2 and Figure 3).

The acquired image datasets were reconstructed into 3D-CT. The tibial tunnel positions were assessed in the medial-to-lateral (ML) and proximal-to-distal (PD) directions of the proximal tibia, as described by Shin et al. [34]. Measurements included the absolute distance from the medial margin of the tibial plateau to the tunnel center in the medial-to-lateral direction and the absolute distance from the posterior margin of the tibial plateau to the tunnel center in the proximal-to-distal direction (Figure 4). To standardize measurement across different knee sizes, relative percentages were computed. This involved dividing the absolute medial-to-lateral distance by the width of the tibial plateau and dividing the absolute proximal-to-distal distance by the distance between the posterior margin of tibial plateau and the champagne glass drop-off.

### 2.5. Patient Demographics, Clinical and Radiologic Measurements, and Complications

The preoperative demographic included age at time of surgery, sex, body mass index (BMI), and concomitant meniscus injury. The clinical and radiologic outcomes and stability were assessed preoperatively and at the 2-year follow-up. The clinical evaluations, including the range of motion (ROM) using a goniometer and knee joint stability classified by the posterior drawer test, were conducted by experienced orthopedic surgeons at the outpatient clinic. The patient-reported outcomes, including the International Knee Documentation Committee (IKDC) subjective score, Lysholm score, and Tegner activity scale, were assessed preoperatively at the time of hospital admission and again at the final follow-up through interviews with the patients by an independent research assistant.

Radiologic measurement included the hip–knee–ankle angle, posterior tibia slope, and the STSD in posterior tibial translation on kneeling stress radiographs. The posterior tibial slope was assessed on true lateral radiographs of the knee using the method originally described by Dejour et al. [35]. The proximal anatomic axis of the tibia was determined by two midpoints between the anterior and posterior tibial cortex, which were located 5 and 10 cm distal to the joint line. A reference line was drawn perpendicular to the proximal anatomic axis of the tibia at the level of the joint line. The posterior tibial slope is defined as the angle between a tangent line connecting to the uppermost anterior and posterior edges of the medial tibial plateau and the reference line.

The measurement of STSD in posterior tibial translation on kneeling stress radiographs, as a main outcome of this study, is illustrated in Figure 5 [36,37]. All radiologic measurements were conducted twice, with a 1-week interval, by two clinical fellows who were blinded to the group allocations.

Postoperative complications, including graft failure, neurovascular injury, compartment syndrome, hemarthrosis, infection, stiffness, and heterotopic calcification were compared between the groups [38,39]. Graft failure was defined by meeting any of the following criteria: (1) the necessity for additional surgery (revision PCL reconstruction, high tibial osteotomy, or arthroplasty) due to unrelieved symptoms, (2) complete graft tear shown on MRI scans, or (3) grade III posterior drawer test or STSD > 10 mm on stress radiographs [40]. Stiffness was identified in patients with loss of >5° of full extension and <120° of knee flexion [41].

### 2.6. Statistical Analyses

Statistical analyses were performed by use of SPSS Version 25.0 (IBM SPSS Statistics; IBM Corp., Armonk, NY, USA). Quantitative variables were presented as mean and SD. For between-group comparisons, the Mann–Whitney U test was applied to compare quantitative variables, while categorical data were compared using the Fisher’s exact test. A *p*-value < 0.05 was regarded as indicative of statistical significance.

A power analysis was performed to determine the sample size required to demonstrate statistical significance. To detect between-group difference in STSD on kneeling stress radiographs, the anticipated STSD on the kneeling stress radiographs of group A and group L was set at 5 mm and 3 mm, respectively, and the standard deviation (SD) was set at 2 mm [40,42]. Alpha was set at 0.05 and the power was set at 0.8. Calculations showed that a minimum sample size of 32 patients (16 patients per group) was required.

To evaluate the interobserver and intraobserver reliability, the intraclass correlation coefficient (ICC) was evaluated for tibial tunnel position and posterior tibial translation measurements. The reliability was defined by ICC values, where values less than 0.5, between 0.5 and 0.75, between 0.75 and 0.9, and greater than 0.90 are indicative of poor, moderate, good, and excellent reliability, respectively [43].

## 3. Results

### 3.1. Preoperative Demographic, Clinical, and Radiologic Data

According to the tibial tunnel positions above or below the champagne glass drop-off on the sagittal view of the proximal tibia, 36 patients were categorized into two groups: anatomic tibial tunnel (group A, *n* = 18) and low tibial tunnel (group L, *n* = 18). The preoperative demographic, clinical, and radiologic data are presented in Table 1.

### 3.2. Tibial Tunnel Position on 3D-CT Scan

In the medial-to-lateral dimension, there was no significant difference in the tibial tunnel positions between the two groups (47.6% in group A vs. 44.5% in group L; *p* = 0.776). Nonetheless, in the proximal-to-distal dimension, the anatomic tibial tunnel was positioned 10.4 mm above the champagne glass drop-off, whereas the low tibial tunnel was 6.7 mm below the champagne glass drop-off. Both the absolute distance and percentage in the proximal-to-distal dimension displayed statistically significant differences between the groups (12.5 mm in group A vs. 29.1 mm in group L, *p* < 0.001; 54.6% in group A vs. 130.6% in group L, *p* < 0.001) (Table 2). Overall, group L exhibited an average 16.6 mm more distal tibial tunnel position without a medial-to-lateral difference compared to group A. The ICC values for interobserver and intraobserver reliability of all tibial tunnel position measurements were greater than 0.75.

### 3.3. Postoperative Outcomes and Complications

At the 2-year follow-up, the patient-reported outcomes and ranges of motion were not significantly different between the two groups. All patients achieved a knee range of motion greater than or equal to 130° flexion without >5° loss to full extension (Table 3). Stability test results, including posterior drawer tests and STSD on kneeling stress radiographs, did not show any statistically significant differences between the groups (2.5 mm in group A vs. 3.7 mm in group L; *p* = 0.346). The ICC values for interobserver and intraobserver reliability were 0.760 and 0.791, respectively, both of which indicated good reliability.

In group A, one patient experienced transient foot drop postoperatively, fully recovering after 3 months of physical therapy, while another patient had surgical wound infection. In group L, one patient presented with hemarthrosis at 6 weeks after surgery. In addition, there was a patient who had incidental findings of a small amount of heterotopic calcification in the popliteal fossa on the CT scan follow-up (Figure 6), without complaining of symptoms or motion loss. No progression of the calcification was noted on plain films during the 2-year-follow-up period. No other complications, such as graft failure, compartment syndrome, or stiffness, were observed in either group. Overall, the incidence of postoperative complications did not show statistically significant differences between the groups (Table 3).

## 4. Discussion

The primary finding of this study was that the clinical outcomes and stability tests were not significantly different between patients who underwent PCL reconstruction with anatomic tibial tunnels and low tibial tunnels. The STSDs on kneeling stress radiographs, the primary outcome of this study, were satisfactory in both groups and did not show significant difference at the 2-year follow-up. Furthermore, there were no significant differences observed in the patient-reported outcomes, including the IKDC subjective score, Lysholm score, and Tegner activity scale, as well as in range of motion, posterior drawer test results, or the incidence of overall complications. To the best of our knowledge, this is the first study comparing clinical outcomes of double-bundle PCL reconstruction using anatomic and low tibial tunnels. Our findings indicate that the clinical and radiologic outcomes of anatomic and low tibial tunnels were parallel at the 2-year follow-up.

Arthroscopic transtibial PCL reconstruction is a challenging procedure compared with ACL reconstruction because of the restricted visualization, the risks of neurovascular injury, and the killer turn effect. The “killer turn” can compromise the posterior stability by repetitive friction between the graft and tunnel inlet, not only attenuating the graft but also enlarging the tunnel inlet, leading to the displacement of the graft [18,44]. Fanelli first described a modified tibial tunnel placement in the inferior lateral part of the PCL fossa to reduce the killer turn effect [25,45,46]. Recent biomechanical studies have shown the impact of the Fanelli tunnel on graft stress and PCL reconstruction laxity. Wang et al. evaluated peak graft stress using a 3D finite element model and found that a specific tibial tunnel placement, 10 mm inferior and 5 mm lateral to the PCL anatomic insertion, resulted in the lowest peak stress on the PCL graft [47]. Another biomechanical study, utilizing 3D-printed tibial models, by Wang et al. demonstrated that low-tibial-tunnel PCL reconstruction significantly reduced stress concentration and graft abrasion compared to anatomic PCL reconstruction [44]. Several studies have explored graft tunnel angles with CT in 2-dimensional planes [48,49,50]. Lin et al. conducted a clinical study quantifying the “killer turn” with 3D-CT in 3D space. They reported that low-tibial-tunnel PCL reconstruction exhibited two significantly gentler turns (superior, 110° and inferior, 151°) compared to the one acute turn (91°) observed in anatomic PCL reconstruction [26].

Despite the initial hypothesis based on previous biomechanical studies, our study surprisingly found no significant difference in clinical and radiologic outcomes between the low tibial tunnel and anatomic tibial tunnel groups in double-bundle PCL reconstruction. A potential explanation for this discrepancy could be rooted in the fact that most prior biomechanical experiments and clinical studies were centered around single-bundle PCL reconstruction. Lately, several randomized controlled studies and systematic reviews with meta-analysis have proposed that double-bundle PCL reconstruction closely restores the native biomechanics of the intact knee and offers superior objective outcomes, such as Tegner activity score, posterior tibial stability on Telos stress radiographs at 90°, and IKDC objective scores, compared to single-bundle reconstruction [11,12,13,14,15]. In other words, the biomechanical advantages associated with double-bundle PCL reconstruction may influence clinical and radiologic outcomes, potentially mitigating the impact of tibial tunnel position.

Another potential reason for the observed discrepancy between results could be that the two tibial tunnel positions in this study did not differ in the medial-to-lateral directions. Various biomechanical studies have indicated that the placement of tibial tunnels in the medial and lateral directions can impact knee laxity [47,49,51]. Galloway et al. conducted a biomechanical study and found that changes in tibial attachment had a minor effect on knee stability, with lateral tibial attachments exhibiting better control of posterior displacement at 30° and 60° flexion compared to medial tibial attachments [51]. In another biomechanical study, by Markolf et al., it was demonstrated that graft forces with a medially placed tibial tunnel were significantly higher than those with a centrally or laterally placed tunnel for flexion angles greater than 65° [52].

This study has some limitations. First, it adopts a retrospective and non-randomized design. Second, given the relatively low incidence of PCL insufficiency, the limited sample size may introduce potential bias to the study. Third, this study lacked an objective evaluation of knee laxity by using arthrometer devices, such as KT-2000 (MEDmetric, San Diego, CA, USA) or GNRB (Genourob, Laval, France), which offer better reproducibility than stress radiographs [53]. Finally, the study provides preliminary short-term clinical outcomes, and further long-term follow-up studies are needed to explore graft survival and late complications, such as osteoarthritis progression.

## 5. Conclusions

The main findings of this study indicated that both anatomic tunnel and low tibial tunnel placements in double-bundle PCL reconstruction demonstrated comparable and satisfactory clinical and radiologic outcomes, with similar overall complication rates at the 2-year follow-up.

## Figures and Tables

**Figure 1 medicina-60-00545-f001:**
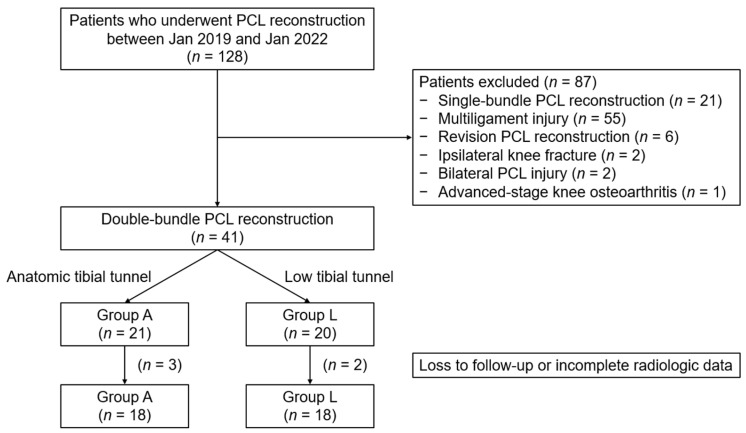
Flowchart of patient enrollment in this study. PCL, posterior cruciate ligament.

**Figure 2 medicina-60-00545-f002:**
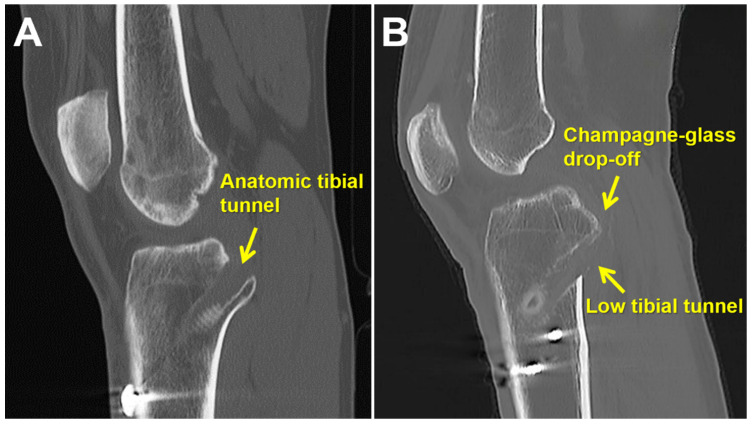
The sagittal view of the proximal tibia computed tomography images showing the position of the tibial tunnel. (**A**) Anatomic tibial tunnel is centered above the champagne glass drop-off. (**B**) Low tibial tunnel is centered below the champagne glass drop-off.

**Figure 3 medicina-60-00545-f003:**
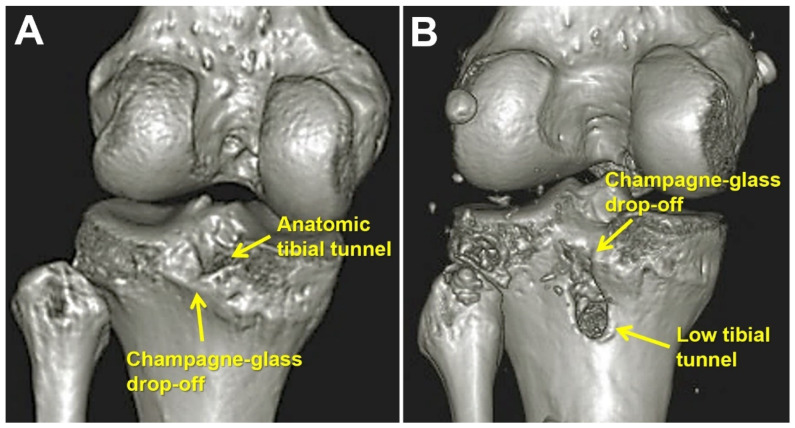
Three-dimensional computed tomography images showing the position of the tibial tunnel. (**A**) Anatomic tibial tunnel; (**B**) low tibial tunnel.

**Figure 4 medicina-60-00545-f004:**
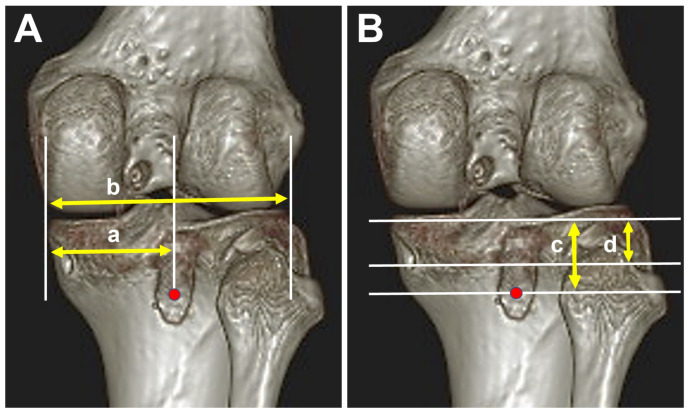
Evaluation of the tibial tunnel position on a 3D-CT scan. The center of the tibial tunnel is marked with a red dot. (**A**) In the medial-to-lateral dimension, a, is distance from the medial margin of tibial plateau to the tibial tunnel center; b, is the total width of the tibial plateau. (**B**) In the proximal-to-distal direction, c, is the distance from the posterior margin of tibial plateau to the center of the tibial tunnel; d, is the distance between the posterior margin of tibial plateau and the champagne glass drop-off.

**Figure 5 medicina-60-00545-f005:**
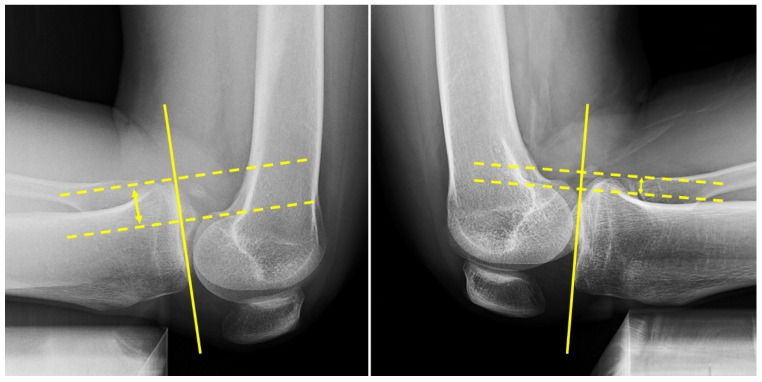
Posterior tibial translation on the kneeling stress radiograph is measured as the distance between the two dotted lines. A continuous line was drawn across the medial tibial plateau, and two dotted lines were drawn perpendicular to the continuous line crossing the midpoint of the femoral condyles and the most posterior border of the tibial plateau. The double arrow indicates the posterior tibial translation.

**Figure 6 medicina-60-00545-f006:**
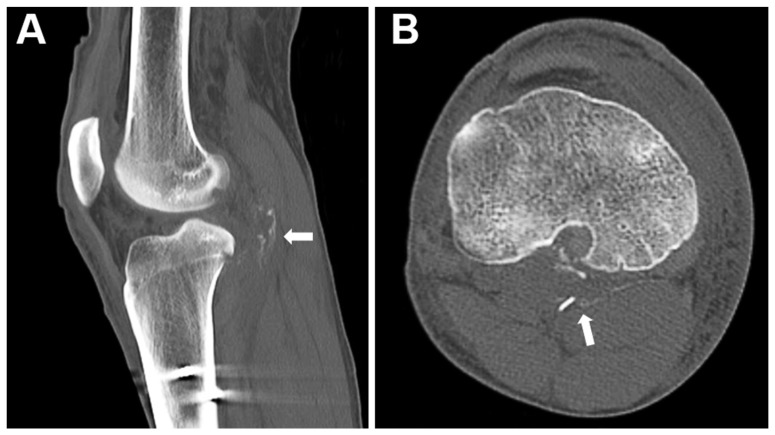
CT scan of the knee demonstrating heterotopic calcification (arrow) in the popliteal fossa on sagittal (**A**) and axial (**B**) planes.

**Table 1 medicina-60-00545-t001:** Preoperative demographic, clinical, and radiologic data.

	Group A (*n* = 18)	Group L (*n* = 18)	*p*-Value
Demographic data			
Age, yr	42.8 ± 9.8	39.0 ± 10.1	0.636
Sex, male: female, *n*	16:2	15:3	0.629
BMI, kg/m^2^	26.7 ± 3.3	24.8 ± 4.8	0.288
Combined meniscus injury, *n* (%)	2 (11%)	5 (28%)	0.402
Clinical data			
ROM, deg	118.6 ± 6.9	119.7 ± 5.2	0.855
Posterior drawer test, grade 0:1:2:3, *n*	0:0:8:10	0:0:7:11	0.753
Radiologic data			
Hip–knee–ankle angle, deg	178.0 ± 2.1	178.4 ± 3.5	0.955
Posterior tibial slope, deg	6.5 ± 2.2	6.4 ± 2.7	0.841
STSD, mm	10.6 ± 2.1	12.5 ± 2.6	0.175
Patient-reported outcomes			
Tegner activity scale	2.8 ± 0.8	2.7 ± 0.8	0.803
IKDC subjective score	49.8 ± 8.5	47.5 ± 9.9	0.642
Lysholm score	62.7 ± 6.7	59.2 ± 8.6	0.393

BMI, body mass index; ROM, range of motion; STSD, side-to-side difference on keeling stress radiographs; IKDC, International Knee Documentation Committee. Data are presented as mean ± standard deviation.

**Table 2 medicina-60-00545-t002:** Tibial tunnel position on 3D-CT scan.

	Group A (*n* = 18)	Group L (*n* = 18)	*p*-Value
ML total, mm	76.8 ± 4.1	76.3 ± 6.7	0.388
ML distance, mm	36.4 ± 7.0	33.9 ± 6.7	0.607
ML percentage, %	47.6 ± 10.0	44.5 ± 4.9	0.776
PD total, mm	22.9 ± 3.6	22.4 ± 2.0	0.768
PD distance, mm	12.5 ± 2.7	29.1 ± 3.5	<0.001 *
PD percentage, %	54.6 ± 6.9	130.6 ± 16.9	<0.001 *

ML, medial to lateral; PD, proximal to distal. * *p*-value < 0.05 indicates statistically significant difference between groups.

**Table 3 medicina-60-00545-t003:** Postoperative outcomes and complications.

	Group A (*n* = 18)	Group L (*n* = 18)	*p*-Value
Clinical and radiologic outcomes			
ROM, deg	134.4 ± 3.9	132.5 ± 4.2	0.340
Posterior drawer test, grade 0:1:2:3, *n*	9:8:1:0	8:8:2:0	0.660
STSD, mm	2.5 ±1.2	3.7 ± 2.0	0.346
Patient-reported outcomes			
Tegner activity scale	4.6 ± 1.2	4.5 ± 1.4	0.816
IKDC subjective score	82.8 ± 8.7	80.5 ± 8.5	0.624
Lysholm score	88.0 ± 9.6	85.8 ± 8.6	0.664
Complications, *n* (%)	2 (11.1)	2 (11.1)	>0.999
Graft failure	0	0	NA
Neurovascular injury	1 (5.6)	0	>0.999
Compartment syndrome	0	0	NA
Hemarthrosis	0	1 (5.6)	>0.999
Infection	1 (5.6)	0	>0.999
Stiffness	0	0	NA
Heterotopic calcification	0	1 (5.6)	>0.999

ROM, range of motion; STSD, side-to-side difference on keeling stress radiographs; IKDC, International Knee Documentation Committee; NA, not applicable.

## Data Availability

Data are contained within the article.

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
