# Peer review of "Anatomic versus Low Tibial Tunnel in Double-Bundle Posterior Cruciate Ligament Reconstruction: Clinical and Radiologic Outcomes with a Minimum 2-Year Follow-Up"

_medicina, 2024, doi:10.3390/medicina60040545_

Round 1

Reviewer 1 Report (New Reviewer)

Comments and Suggestions for Authors

Comments on the Quality of English Language

None.

Author Response

Comments

Response

Reviewer 1

1.  Did this study support the non-anatomical PCLR instead of anatomical PCLR? There is an evidence that anatomical PCLR is superior to non-anatomical PCLR (Winkler et al, KSSTA, 2021). In fact, Fanelli et al reported the importance of anatomic tunnel positions during PCLR in 2020 (Sports Med Arthrosc Rev. 2020 Mar;28(1):8-13.doi: 10.1097/JSA.0000000000000255.)

Thank for your comment. We attempted to improved our outcomes of DB-PCLR. Therefore, we performed DB-PCLR using low tibial tunnel in light of numerous biomechanic studies revealed this technique had an advantage to mitigate the “killer turn” effect. In fact, Yoon et al reported comparable clinical outcomes between anatomic and low tibial tunnel in remnant-preserving PCLR, which was close to our results. (Orthop J Sports Med. 2021 Feb. doi: 10.1177/2325967120985153. PMID: 33709007)

2.  Was the PCLR using a low tibial tunnel performed intentionally or accidentally?

Thank for your comment. All of the DB-PCLR with a low tibial tunnel were performed intentionally.

3.  If the authors agree with the PCLR with a low tibial tunnel, why not perform a prospective study?

Thank for your comment. While a prospective study has its advantages in establishing causality and minimizing biases, we opted for a retrospective study due to considerations of feasibility and practicality, which served as an initial exploration or hypothesis-generating phase to assess the feasibility and potential outcomes before committing to a larger-scale prospective study.

4.  Patient Reported Outcome Measures such as IKDC, Lysholm, KOOS and Tegner are not reported, limiting the understanding of the postoperative clinical outcomes. The authors reported that there was no failure case in the present study. However, the prevalence of failure after PCLR has been reported to be 10-25%. Therefore, no case of failure after surgery may be due to insufficient reporting of the surgical outcomes.

Thank you for the recommendation.

We incorporated the pre- and post-operative PROMs, including the Tegner scale, subjective IKDC, and Lysholm score, in the revised manuscript (Table 1). In this study, a total of 3 patients (8.3%) presented with Gr.II posterior drawer test at the 2-year follow-up. However, all of the 3 patients had the STSD <10mm on kneeling stress view, could tolerate minor symptoms, and were not considered to have revision surgery yet. Thus, they did not meet the definition of graft failure in this preliminary study.

5.  The interrater and intrarater reliabilities of the radiological findings need to be confirmed.

Thank you for the recommendation.

Interobserver and intraobserver reliability for radiologic measurements was added in 2.6. Statistical Analysis.

6.  How many isolated PCLR were performed in the study period? Please show the flowchart of the study.

Thanks for the comment. We explained the patient enrollment in the revised 2.1. Patient Enrollment and Study Design with providing a Flowchart (Figure 1).

Line 37, 38: Because of a retrospective study, the definitive conclusion cannot be drawn. Please revise these sentences.

Thanks for the comment. Conclusion was revised as “the main findings of this study indicated that both anatomic tunnel and low tibial tunnel placements in double-bundle PCL reconstruction demonstrated comparable and satisfactory clinical and radiologic outcomes, with similar overall complication rates at the 2-year follow-up.” (Line 335-338)

Line 52-54: These previous studies have compared anatomical SB- vs DB-PCLR. Thus, anatomical PCLR is the fundamental concept of the discussion.

Thank you for the recommendation. The “anatomic-based” DB-PCLR was added.

Line 73-83:

What was the inclusion criteria of the study?

Patients who underwent double-bundle PCL reconstruction in our hospital between Jan 2019 and Jan 2022 were included in this study (Line 90).

Advanced-stage knee osteoarthritis? How did the authors define this?

Thank you for pointing this out. We defined advanced-staged knee osteoarthritis as Kellgren-Lawrence grade 3 or 4 in this study, which was added in the revised draft (Line 100-101).

Incomplete follow-up? Please describe this more clearly.

Patients had lost to follow-up before the end of 2-year period (Line 177).

Line 92-95: Why did the authors perform PCLR using a low tibial tunnel?  How did the authors determine the surgical selection of anatomic vs non-anatomic PCLR?

Thanks for the comment. In response to above 1, we performed PCLR using a low tibial tunnel, attempting to improve our outcomes. Because the clinical effect of low tibial tunnel was unclear, the preferred patient selection for this technique was young, high-activity, lower BMI, and greater pre-op STSD. However, the final surgical selection was with respect to each surgeon’s decision, and the pre-op demographic data did not show statistically significant differences.

How many orthopaedic surgeons performed PCLRs in the study period?

There were 3 orthopedic surgeons performed DB-PCLRs in this study.

Line 96-107: Did the authors create the 9mm tunnel of ALB and 8mm tunnel of PMB for all patients? How about female patients who have a smaller knee?

Thank you for pointing this out.

The size of femroal attachments of the

AMB and PMB were broader than its

tibial attachemnt, ranging 112-118mm2

and 60-90mm2 respectively. Thus, in our

practice, we aimed to create th 9mm and

8mm tunnels for ALB and PMB

respectively to make up a total 12mm

width PCL graft ideally. If the patient had

a small-size knee, 1mm-adjustment for

the size of tunnel can be tolerated.

(LaPrade, C.M., et al., Emerging

Updates on the Posterior Cruciate

Ligament: A Review of the Current

Literature. Am J Sports Med, 2015.

43(12): p. 3077-92.)

Line 109-116: Did the authors change the postoperative rehabilitation protocols depending on the concomitant meniscal tears?

In our PCLR Postoperative Rehabilitation protocol, limited knee flexion <90° was restricted for the initial 4 weeks and unrestricted range of motion was allowed after 6 weeks. This protocol is consistent with our current rehab protocol for meniscal repair.

Line 159, 160: Please evaluate the reliability of the study findings.

Thank you for pointing this out. Interobserver and intraobserver reliability for posterior tibial translation measurements was added in 2.6. Statistical Analysis (Line 223-227).

Table 1:

Tegner activity scale needs to be added.

Pre-and post-operative Tegner activity scale was added.

Please show the reliability of the radiologic data. How did the authors evaluate the tibial slope?

The posterior tibial slope was assessed on true lateral radiographs of the knee following the methodology originally outlined by Dejour et al., as illustrated upon in the revised manuscript. The ICC values for interobserver and intraobserver reliability were 0.784 and 0.853, respectively.

[Dejour H, Bonnin M. Tibial translation after anterior cruciate ligament rupture. Two radiological tests compared. J Bone Joint Surg Br. 1994;76(5):745-749.]

Table 2:

Please evaluate the reliability of the study findings.

The ICC values for interobserver and intraobserver reliability were 0. 866 and 0. 907, respectively (Line 244-246).

Reviewer 2

Line 58: add the reference number [13] as the references are listed by numbers and not in alphabetical order.

Thanks for the comment. We have rearranged our reference order in the revised manuscript.

Did the authors obtain ethical approval?

This study had been approved by the Institutional Review Board of Chang Gung Medical Foundation (reference No. IRB 202400227B0) and the authors obtained the informed consent from the patients to publish this paper.

Reviewer 2 Report (New Reviewer)

Comments and Suggestions for Authors

Please see attached

Comments on the Quality of English Language

No major issues with the English language. The structure, writing and flow are very good.

Author Response

Comments

Response

Reviewer 1

1.  Did this study support the non-anatomical PCLR instead of anatomical PCLR? There is an evidence that anatomical PCLR is superior to non-anatomical PCLR (Winkler et al, KSSTA, 2021). In fact, Fanelli et al reported the importance of anatomic tunnel positions during PCLR in 2020 (Sports Med Arthrosc Rev. 2020 Mar;28(1):8-13.doi: 10.1097/JSA.0000000000000255.)

Thank for your comment. We attempted to improved our outcomes of DB-PCLR. Therefore, we performed DB-PCLR using low tibial tunnel in light of numerous biomechanic studies revealed this technique had an advantage to mitigate the “killer turn” effect. In fact, Yoon et al reported comparable clinical outcomes between anatomic and low tibial tunnel in remnant-preserving PCLR, which was close to our results. (Orthop J Sports Med. 2021 Feb. doi: 10.1177/2325967120985153. PMID: 33709007)

2.  Was the PCLR using a low tibial tunnel performed intentionally or accidentally?

Thank for your comment. All of the DB-PCLR with a low tibial tunnel were performed intentionally.

3.  If the authors agree with the PCLR with a low tibial tunnel, why not perform a prospective study?

Thank for your comment. While a prospective study has its advantages in establishing causality and minimizing biases, we opted for a retrospective study due to considerations of feasibility and practicality, which served as an initial exploration or hypothesis-generating phase to assess the feasibility and potential outcomes before committing to a larger-scale prospective study.

4.  Patient Reported Outcome Measures such as IKDC, Lysholm, KOOS and Tegner are not reported, limiting the understanding of the postoperative clinical outcomes. The authors reported that there was no failure case in the present study. However, the prevalence of failure after PCLR has been reported to be 10-25%. Therefore, no case of failure after surgery may be due to insufficient reporting of the surgical outcomes.

Thank you for the recommendation.

We incorporated the pre- and post-operative PROMs, including the Tegner scale, subjective IKDC, and Lysholm score, in the revised manuscript (Table 1). In this study, a total of 3 patients (8.3%) presented with Gr.II posterior drawer test at the 2-year follow-up. However, all of the 3 patients had the STSD <10mm on kneeling stress view, could tolerate minor symptoms, and were not considered to have revision surgery yet. Thus, they did not meet the definition of graft failure in this preliminary study.

5.  The interrater and intrarater reliabilities of the radiological findings need to be confirmed.

Thank you for the recommendation.

Interobserver and intraobserver reliability for radiologic measurements was added in 2.6. Statistical Analysis.

6.  How many isolated PCLR were performed in the study period? Please show the flowchart of the study.

Thanks for the comment. We explained the patient enrollment in the revised 2.1. Patient Enrollment and Study Design with providing a Flowchart (Figure 1).

Line 37, 38: Because of a retrospective study, the definitive conclusion cannot be drawn. Please revise these sentences.

Thanks for the comment. Conclusion was revised as “the main findings of this study indicated that both anatomic tunnel and low tibial tunnel placements in double-bundle PCL reconstruction demonstrated comparable and satisfactory clinical and radiologic outcomes, with similar overall complication rates at the 2-year follow-up.” (Line 335-338)

Line 52-54: These previous studies have compared anatomical SB- vs DB-PCLR. Thus, anatomical PCLR is the fundamental concept of the discussion.

Thank you for the recommendation. The “anatomic-based” DB-PCLR was added.

Line 73-83:

What was the inclusion criteria of the study?

Patients who underwent double-bundle PCL reconstruction in our hospital between Jan 2019 and Jan 2022 were included in this study (Line 90).

Advanced-stage knee osteoarthritis? How did the authors define this?

Thank you for pointing this out. We defined advanced-staged knee osteoarthritis as Kellgren-Lawrence grade 3 or 4 in this study, which was added in the revised draft (Line 100-101).

Incomplete follow-up? Please describe this more clearly.

Patients had lost to follow-up before the end of 2-year period (Line 177).

Line 92-95: Why did the authors perform PCLR using a low tibial tunnel?  How did the authors determine the surgical selection of anatomic vs non-anatomic PCLR?

Thanks for the comment. In response to above 1, we performed PCLR using a low tibial tunnel, attempting to improve our outcomes. Because the clinical effect of low tibial tunnel was unclear, the preferred patient selection for this technique was young, high-activity, lower BMI, and greater pre-op STSD. However, the final surgical selection was with respect to each surgeon’s decision, and the pre-op demographic data did not show statistically significant differences.

How many orthopaedic surgeons performed PCLRs in the study period?

There were 3 orthopedic surgeons performed DB-PCLRs in this study.

Line 96-107: Did the authors create the 9mm tunnel of ALB and 8mm tunnel of PMB for all patients? How about female patients who have a smaller knee?

Thank you for pointing this out.

The size of femroal attachments of the

AMB and PMB were broader than its

tibial attachemnt, ranging 112-118mm2

and 60-90mm2 respectively. Thus, in our

practice, we aimed to create th 9mm and

8mm tunnels for ALB and PMB

respectively to make up a total 12mm

width PCL graft ideally. If the patient had

a small-size knee, 1mm-adjustment for

the size of tunnel can be tolerated.

(LaPrade, C.M., et al., Emerging

Updates on the Posterior Cruciate

Ligament: A Review of the Current

Literature. Am J Sports Med, 2015.

43(12): p. 3077-92.)

Line 109-116: Did the authors change the postoperative rehabilitation protocols depending on the concomitant meniscal tears?

In our PCLR Postoperative Rehabilitation protocol, limited knee flexion <90° was restricted for the initial 4 weeks and unrestricted range of motion was allowed after 6 weeks. This protocol is consistent with our current rehab protocol for meniscal repair.

Line 159, 160: Please evaluate the reliability of the study findings.

Thank you for pointing this out. Interobserver and intraobserver reliability for posterior tibial translation measurements was added in 2.6. Statistical Analysis (Line 223-227).

Table 1:

Tegner activity scale needs to be added.

Pre-and post-operative Tegner activity scale was added.

Please show the reliability of the radiologic data. How did the authors evaluate the tibial slope?

The posterior tibial slope was assessed on true lateral radiographs of the knee following the methodology originally outlined by Dejour et al., as illustrated upon in the revised manuscript. The ICC values for interobserver and intraobserver reliability were 0.784 and 0.853, respectively.

[Dejour H, Bonnin M. Tibial translation after anterior cruciate ligament rupture. Two radiological tests compared. J Bone Joint Surg Br. 1994;76(5):745-749.]

Table 2:

Please evaluate the reliability of the study findings.

The ICC values for interobserver and intraobserver reliability were 0. 866 and 0. 907, respectively (Line 244-246).

Reviewer 2

Line 58: add the reference number [13] as the references are listed by numbers and not in alphabetical order.

Thanks for the comment. We have rearranged our reference order in the revised manuscript.

Did the authors obtain ethical approval?

This study had been approved by the Institutional Review Board of Chang Gung Medical Foundation (reference No. IRB 202400227B0) and the authors obtained the informed consent from the patients to publish this paper.

Round 2

Reviewer 1 Report (New Reviewer)

Comments and Suggestions for Authors

Comments on the Quality of English Language

None.

Author Response

Comment 1:

        In the Patient Enrollment and Study Design section, please describe the indications for low-tibial tunnel PCLR.

Response:

Revised accordingly. Thanks for your comment.

Comment 2:

        Line 68-73: The high tibial osteotomy to steepen the posterior tibial slope and creating a tibial tunnel with a larger angle have been reported to be a promising technique for mitigating the killer-turn effect during PCLR. Please consider citing a paper by Yang F et al. (Influence of the Tibial Tunnel Angle and Posterior Tibial Slope on "Killer Turn" during Posterior Cruciate Ligament Reconstruction: A Three-Dimensional Finite Element Analysis. J Clin Med. 2023 Jan 19;12(3):805. doi: 10.3390/jcm12030805. )

Response:

        Thank you for providing this finite element study, which demonstrated the influence of the anterior opening wedge HTO (aOW-HTO) and tibial tunnel angles on the stress distribution in the “killer turn” of the PCL graft. We have added the reference in the revised manuscript (Line 70).

Comment 3:

        Line 246-248: This study evaluated six parameters of the tibial tunnel position on 3D-CT. Therefore, please show the intra- and inter-rater reliability in each parameter. In addition, 95%CI needs to be shown with ICC values.

Response:

        Thank for your kind reminders. Actually, there were 4 independent parameters in Table 2 (ML and PD relative percentages were calculated according to the knee size and the tunnel position). The ICC values for inter-and intra-observer reliability for ML total, ML distance, PD total, PD distance were 0.784/0.914, 0.760/0.853, 0.815/0.766, and 0.866/0.907 respectively. All ICC values were >0.75.

Comment 4:

        In the postoperative rehabilitation section, please add the comments that postoperative rehabilitation protocol was the same regardless the intervention to meniscal tears.

Response:

Revised accordingly. Thanks for your comment.

This manuscript is a resubmission of an earlier submission. The following is a list of the peer review reports and author responses from that submission.

Round 1

Reviewer 1 Report

Comments and Suggestions for Authors

Dear authors,

Dear editorial board,

thank you for submitting this manuscript and the opportunity to review it.

Overall, this manuscript is well-written. However, the concerns outweigh the relevance of this paper. Such a small sample size does not allow for any statistical comparisons. The conclusion that both tunnel positions had comparable outcomes especially lacks sufficient reflection of their data.  All analyses shown are nonsense. We cannot conclude comparable outcomes in such small samples based on insignificant p-values. I'm afraid that's not right.

I strongly recommend that the authors not publish this manuscript without a sufficiently large cohort. Maybe they need to prolong the study period to achieve a sufficiently large cohort. Additionally, I would strongly recommend a sample size calculation a priori.

Comments on the Quality of English Language

Quality of English overall fine

Author Response

Thanks for the comment. A power analysis was performed to determine the sample size required to demonstrate statistical significance. To detect between-group difference in STSD on kneeling stress radiographs, the anticipated STSD on kneeling stress radiographs of group A and group L was set at 5 mm and 3 mm respectively and the standard deviation (SD) was set at 2 mm [27, 29]. Alpha was set at 0.05, and the power was set at 0.8. Calculations showed a minimum sample size of 32 patients (16 patients per group) was required. Therefore, we increased the inclusion date from May 2020 to December 2021, to Jan 2019 and Jan 2022. And we enrolled more patients in the cohort (previously, 5 patients in Group A and 5 in Group L; to 18 patients in Group A and 18 in Group L). Now, we had 18 patients in Group A and 18 patients in Group L for analysis.

Reviewer 2 Report

Comments and Suggestions for Authors

This is a very interesting paper on PCL reconstruction. Killer turn have been proposed as one of the major factors in PCLR failure. Therefore attempts in orthopaedic society have been made to overcome this issue including eg onlay technique. This paper is soundly prepared and written, however, very small study group gives a great bias to the results. I understand that incidence of PCL I limited, nevertheless I believe that this sample size is to small to conclude anything on it. Therefore I would recommend to gain some more patients in the study group maybe in collaboration with other hospitals and then prepare a paper supporting adequate sample size.

Comments on the Quality of English Language

This is a very interesting paper on PCL reconstruction. Killer turn have been proposed as one of the major factors in PCLR failure. Therefore attempts in orthopaedic society have been made to overcome this issue including eg onlay technique. This paper is soundly prepared and written, however, very small study group gives a great bias to the results. I understand that incidence of PCL I limited, nevertheless I believe that this sample size is to small to conclude anything on it. Therefore I would recommend to gain some more patients in the study group maybe in collaboration with other hospitals and then prepare a paper supporting adequate sample size.

Author Response

(The authors gave the same response as above.)
